# Research on Hydrodynamic Characteristics of Electronic Paper Pixels Based on Electrowetting

**DOI:** 10.3390/mi14101918

**Published:** 2023-10-10

**Authors:** Mingzhen Chen, Shanling Lin, Ting Mei, Ziyu Xie, Jianpu Lin, Zhixian Lin, Tailiang Guo, Biao Tang

**Affiliations:** 1National and Local United Engineering Laboratory of Flat Panel Display Technology, School of Physics and Information Engineering, Fuzhou University, Fuzhou 350108, China; 221110007@fzu.edu.cn (T.M.); 211120046@fzu.edu.cn (Z.X.); gtl@fzu.edu.cn (T.G.); 2School of Advanced Manufacturing, Fuzhou University, Quanzhou 362200, China; sllin@fzu.edu.cn (S.L.); ljp@fzu.edu.cn (J.L.); 3Guangdong Provincial Key Laboratory of Optical Information Materials and Technology, South China Academy of Advanced Optoelectronics, South China Normal University, Guangzhou 510006, China; tangbiao@scnu.edu.cn

**Keywords:** electrowetting on dielectric (EWOD), photoelectric characteristics, photoelectric response, electronic paper, micro-hydrodynamics, fluid viscosity, driving waveform

## Abstract

In this paper, we propose a driving waveform with a complex ramp pulse for an electrowetting display system. The relationship between the contact angle and viscosity of inks was calculated based on the fluid-motion characteristics of different viscosities. We obtained the suitable range of viscosity and voltage in the liquid–oil–solid three-phase contact display system. We carried out model simulation and driving waveform design. The result shows that the driving waveform improves the response speed and aperture ratio of electrowetting. The aperture ratio of electrowetting pixels is increased to 68.69%. This research is of great significance to optimizing the structure of fluid material and the design of driving waveforms in electrowetting displays.

## 1. Introduction

With the continuous development of electronic paper technology, it can be divided into cholesteric displays and electrophoretic displays according to the material. These two types of electronic paper have the advantages of dual stability and low power consumption. However, it is difficult to achieve dynamic display using these technologies, and severe remnants are easily generated because of their slow response [1,2,3,4]. On the contrary, because electrowetting displays have fast response, low power consumption, easy colorization, and other advantages the technology of electrowetting displays has led to a research upsurge in reflective dynamic video displays. During the research process, researchers mainly conduct basic research on ink and solution materials, pixel structures, device processes, physical mechanisms, and photoelectric response characteristics. However, due to the influence of electrowetting viscosity and device defects, there is still the problem of low aperture ratio caused by ink breakage and backflow. It is significant to research the relationship between the viscosity of electrowetting and its photoelectric properties.

As early as 1872, Gabriel Lippmann found that the surface tension would change the meniscus of mercury solution when the mercury solution with a voltage difference was covered by a dilute sulfuric acid in the Kirchhoff laboratory in Heidelberg, and the theory of electrocapillary effect was put forward [5]. In 1981, Beni researched the micro-hydrodynamics of electrowetting devices in Bell laboratory in Brazil [6]. After discovering that the devices had memory and voltage threshold characteristics, he proposed a voltage-driven display scheme with matrix addressing. Restolh et al. found that the relationship between contact angle and positive and negative voltage presented symmetrical curve characteristics when low voltage was applied [7]. It was usually expressed by the Young–Lippmann equation before contact angle saturation. When the applied voltage reached the threshold voltage, the contact angle of the ink reached saturation. At this time, when the voltage was reduced, the contact angle did not return to its initial value. It resulted in contact angle hysteresis, which affected the response speed of the ink. This phenomenon also occurred at low voltage, usually caused by the charge capture at the solid–liquid interface. Using ink-jet printing technology, Ku et al. employed electrodes of varying shapes in electrowetting to simulate the alteration of microfluidics [8].

Hsieh et al. studied the optoelectronic performance of ink contact angles by deriving pixel hydrodynamic behavior from a Maxwell stress tensor and the flow phase field of the water–oil layer [9]. From an electrochemical perspective, Rivas et al., found that the degree of contact angle deformation could be controlled by increasing the solvation energy difference in order to suppress the contact angle saturation with an applied electric field [10,11,12,13,14,15]. With the electrostatic force characteristics of the capillary, the charge on the three-phase contact wire constantly accumulated, producing a certain electrostatic force, which caused the capillary tension to change. Thus, it made the contact angle of the ink change and led to the hysteresis of the contact angle [16,17,18,19,20]. Accordingly, it is important to study the influence of ink with different viscosities on the photoelectric response characteristics.

In addition, some researchers have found that electrowetting viscosity can affect fluid motion but have not conducted correlation studies with hysteretic contact angle and low aperture ratio of electrowetting pixels [21,22,23]. To address these issues in electrowetting, researchers have proposed a series of driving waveform studies [24]. A DC and AC driving waveform was proposed to suppress ink backflow and breakage [25]. In order to improve the response of electrowetting, an equivalent driving waveform was designed with an overdriving voltage [26]. For the sake of improving the aperture ratio of electrowetting, a driving waveform with a narrow descent slope, low voltage maintenance, and ascent slope was proposed to suppress ink backflow [27]. Some people also proposed a combination waveform with an ascending gradient to improve the stability and suppress charge capture [28]. It can be seen that different driving waveforms have different effects on the electrowetting motion. Therefore, based on the influence of electrowetting viscosity on the motion and photoelectric characteristics, it is of great significance that a suitable driving waveform is proposed to improve the electrowetting display.

In this paper, the photoelectric characteristic model of electrowetting ink with different viscosities is established from the perspective of electrodynamics under different driving voltage conditions. And the causes of different characteristics of different viscosities of ink are analyzed. Additionally, a compound driving waveform is designed to improve the response speed of the oil film and suppress oil film rupture, ink backflow, and charge capture. The relevant experimental testing platform was built to test the photoelectric characteristics of motion response with two kinds of viscosity inks and applied to driving voltages of different frequencies and amplitudes. Furthermore, the feasibility of the model was verified with the driving effect of the composite waveform. Ultimately, the experimental results are summarized and evaluated.

## 2. Principle Analysis and Mathematical Modeling

Electrowetting electronic paper is a reflective display based on microfluidics. Its single-pixel structure is composed of a transparent cover plate, conductive solution, chromatic insulating inks, pixel walls of both sides, a supported column, a hydrophobic layer, and a white reflection base. In Figure 1a, the electrowetting pixel will flush the entire pixel without applied voltage. In Figure 1b, when a voltage is applied, the ink will shrink at different rates and to different degrees, and the white reflective substrate is revealed.

After the voltage is applied, the ink of the electrowetting shrinks in pixels. The area covered by the ink is approximately considered as a combination of many small sectors, and its white aperture ratio (WAR) is calculated as:(1)$$\mathrm{WAR}=1-\int_{-\pi }^{\pi }R^{2}(\alpha )d\alpha /2l^{2}\times 100\%,\;0\leq R\leq l/\sqrt{2}$$
where a pixel center is taken as the origin O of the polar coordinates, α is the horizontal polar coordinate included angle of ink shrinkage, Δα indicates the angle of the approximate small sector, $R\left(\alpha \right)$ represents the side length of the sector in polar coordinates, and l is the side length of the square pixel. The range of aperture ratio when the oil shrinks is shown in Figure 2 in electrowetting pixels.

According to the Young–Lippmann equation, the relationship between the three-phase contact angle and the droplet surface tension is as follows [11]:(2)$$\gamma_{LO}\cos\theta =\gamma_{SO}-\gamma_{SL}$$
where $\gamma_{LO}$ is the liquid–oil contact surface; $\gamma_{SO}$ is the solid–oil contact surface; $\gamma_{SL}$ is the solid–liquid contact surface; and θ is the oil–solid–liquid three-phase contact angle.

After the voltage is applied to the electrowetting pixel, the total potential energy E of the electrowetting includes the potential energy of the oil–liquid contact surface $E_{OL}$, the potential energy of the bottom surface $E_{OS}$, the electric field potential energy $E_{e}$, and the dynamic friction potential energy of the liquid–oil and solid–oil contact surfaces $E_{fOL}$ and $E_{fOS}$, namely
(3)$$E=E_{OL}+E_{OS}+E_{e}-E_{fOL}-E_{fOS}$$

The potential energy of oil–liquid interface is related to the oil–liquid interface and the contact area, which can be expressed as
(4)$$E_{OL}=\gamma_{OL}S_{OL}=\gamma_{OL}k_{0}(\cos\theta_{0}-\cos\theta )\int_{-\pi }^{\pi }R^{2}(\alpha )d\alpha$$
where $\gamma_{OL}$ is the interfacial tension between the ink and the solution, $S_{OL}$ is the contact area of the ink and the solution, $k_{0}$ is the correlation coefficient of the oil surface, $\theta_{0}$ is the contact angle of the oil film at zero voltage, and θ is the current contact angle.

Similarly, the potential energy on the bottom can be expressed in terms of oil–solid interface tension, and the contact area as follows:(5)$$E_{OS}=\gamma_{OS}S_{OS}=\gamma_{OS}\int_{-\pi }^{\pi }R^{2}(\alpha )d\alpha$$
where $\gamma_{OS}$ is the interfacial tension between the ink and the hydrophobic dielectric layer, and $S_{OS}$ is the contact area between the ink and the hydrophobic dielectric layer.

We regard the bottom surface, hydrophobic dielectric layer, and electrode plate as a parallel-plate capacitor, and its potential energy of the electric field is
(6)$$E_{e}=CU^{2}/2=\varepsilon_{0}\varepsilon_{1}(S_{OS})/d_{f}\cdot U^{2}/2=\varepsilon_{0}\varepsilon_{1}U^{2}\int_{-\pi }^{\pi }R^{2}(\alpha )d\alpha /2d_{f}$$
where C is the capacitance, U is the driving voltage; $\varepsilon_{0}$ and $\varepsilon_{1}$ are the vacuum dielectric constant and relative dielectric constant, respectively; and $d_{f}$ is the thickness of the dielectric layer.

The potential energy of the dynamic friction on the liquid–oil contact surface is
(7)$$E_{fOL}=(1/k_{0}+k_{1}v_{OL}^{m})S_{OL}=(1+k_{0}k_{1}v_{OL}^{m})(\cos\theta_{0}-\cos\theta )\int_{-\pi }^{\pi }R^{2}(\alpha )d\alpha$$
where $k_{0}$ is the capillary coefficient, $k_{1}$ is the correlation coefficient between the friction and the moving speed on the liquid–oil surface, and $v_{OL}$ is the moving speed on the liquid–oil surface.

The potential energy of the dynamic friction on the oil–solid contact surface is
(8)$$E_{fOS}=k_{2}v_{OS}^{n}S_{OS}=k_{2}v_{OS}^{n}\int_{-\pi }^{\pi }R^{2}(\alpha )d\alpha$$
where $k_{2}$ is the correlation coefficient between the friction and the moving speed on the liquid–oil contact surface, and $v_{OS}$ is the moving speed of the ink on the hydrophobic dielectric layer.

When the electrowetting reaches equilibrium, the electrowetting meets the following relationship:(9)$$E_{OS}+E_{e}+E_{OL}-E_{fOL}-E_{fOS}=0$$

Equations (4)–(8) are substituted into Equation (9) to obtain the contact angle, as follows:(10)$$\cos\theta =\cos\theta_{0}-\frac{\varepsilon_{0}\varepsilon_{1}U^{2}/2d_{f}+\gamma_{OS}-k_{2}v_{OS}^{n}}{1+k_{0}k_{1}v_{OL}-k_{0}\gamma_{OL}}$$
where $\varepsilon_{0}$ and $\varepsilon_{1}$ are the vacuum permittivity and relative permittivity, respectively; U is the applied voltage; $d_{f}$ is the thickness of the dielectric layer; $k_{0}$ is the capillary coefficient; $k_{1}$ is the correlation coefficient between the friction of the liquid–oil contact surface and the moving speed of the contact line; $k_{2}$ is the correlation coefficient between the friction of the solid–oil contact surface and the moving speed of the contact line; $v_{OS}$ is the moving speed of the ink on the hydrophobic dielectric layer; and $v_{OL}$ is the moving speed of the ink contact line.

Because ink viscosity will affect the interfacial tension of liquid–oil contact surface and oil–solid contact surface, it can be expressed as
(11)$$\eta =\gamma_{OL}d_{OL}/v_{OL}S_{OL}=\gamma_{OS}d_{OS}/v_{OS}S_{OS}$$
where η is the viscosity of the ink (unit: Pa·s); $d_{OL},d_{OS}$ are the displacement of the oil film on the contact line and the hydrophobic layer, respectively; and $v_{OL},v_{OS}$ are the movement speed of the oil film on the contact line and the hydrophobic layer, respectively.

Based on Equations (10) and (11), the contact angle relationship from the movement to the hysteresis is
(12)$$\cos\theta =\cos\theta_{0}-\frac{\varepsilon_{0}\varepsilon_{1}U^{2}/2d_{f}+\eta v_{OS}S_{OS}/d_{OS}-k_{2}v_{OS}^{n}}{1+k_{0}k_{1}v_{OL}^{m}-k_{0}\eta v_{OL}S_{OL}/d_{OL}}$$

The viscosity of the ink and the rough surface of the hydrophobic layer affect its response speed and movement amplitude. At the same time, the hysteresis of the contact angle is also caused by charge capture. Therefore, from the movement of the applied voltage ink to the contact angle saturation hysteresis, the contact angle relationship is
(13)$$\cos\theta =\left\{\begin{array}{l} \cos\theta_{0}-\varepsilon_{0}\varepsilon_{1}U^{2}/2d_{f},\;0\leq U<U_{th1}\\ \cos\theta_{0}-\frac{\varepsilon_{0}\varepsilon_{1}U^{2}/2d_{f}+\eta v_{OS}S_{OS}/d_{OS}-k_{2}v_{OS}^{n}}{1+k_{0}k_{1}v_{OL}^{m}-k_{0}\eta v_{OL}S_{OL}/d_{OL}},U_{th1}\leq U<U_{th2}\\ \cos\theta_{1},\;U\geq U_{th2} \end{array}\right.$$

In addition, in order to avoid movement fracture and contact angle hysteresis of the oil film, the sufficient potential energy of the liquid–oil interfacial tension prevents oil droplets from escaping, which follows the following rules:(14)$$\gamma_{OS}\;\leq \;\gamma_{O}=\eta v_{O}S_{O}/d_{O}\;\leq \;\gamma_{OL}$$
where $\gamma_{O}$ is the relative static friction in the molecular layer inside the ink, and $v_{O},d_{O},S_{O}$ are the relative velocity, the relative displacement, and the contact area in the ink molecular layer, respectively.

According to Equation (14), the suitable viscosity range of electrowetting is
(15)$$\gamma_{OS}d_{O}/v_{O}S_{O}<\eta <\gamma_{OL}d_{O}/v_{O}S_{O}$$

When the oil film of electrowetting begins to shift with the applied voltage $U_{th1}$, the electric field driving force $F_{e}$ is greater than the sum of the maximum static friction $F_{\mathrm{OL}}$ of the ink interface layer and the capillary force $F_{\mathrm{SOL}}$ of the three-phase contact, namely $F_{e}\geq F_{\mathrm{OL}}+F_{\mathrm{SOL}}$. The specific expression is
(16)$$\left(\gamma_{OL}+\gamma_{OS})\varepsilon_{0}\varepsilon_{1}U^{2}/2d_{f}\;\geq \;\gamma_{OL}\varepsilon_{0}\varepsilon_{1}U^{2}/2d_{f}+\eta v_{0}\gamma_{OL}(\cos\theta_{0}-\cos\theta \right)$$

From Equation (16), the initial voltage threshold $U_{th1}$ for the electrowetting movement is
(17)$$\;U_{th1}=\sqrt{2d_{f}\eta v_{0}\gamma_{OL}(\cos\theta_{0}-\cos\theta )/\gamma_{OS}\varepsilon_{0}\varepsilon_{1}}$$

When the applied voltage reaches $U_{th2}$, a part of the electric charge will be trapped in the hydrophobic dielectric layer, which results in the contact angle hysteresis. With the further increase in voltage, there will still be a large number of ions entering the oil film. The energy exceeds the surface potential energy of the oil film, causing the oil film to burst. At this time, the electric field driving force $F_{e}$ is greater than the sum of the maximum static friction $F_{\mathrm{OL}}$ of the ink interface layer, the capillary force $F_{\mathrm{SOL}}$ of three-phase contact, and the bottom static friction $F_{\mathrm{SO}}$, that is $F_{e}\geq F_{\mathrm{OL}}+F_{\mathrm{SOL}}+F_{\mathrm{SO}}$. The specific expression is
(18)$$(\gamma_{OL}+\gamma_{OS})\varepsilon_{0}\varepsilon_{1}U^{2}/2d_{f}\geq \;\gamma_{OL}\varepsilon_{0}\varepsilon_{1}U^{2}/2d_{f}+\eta v_{1}\gamma_{OL}(\cos\theta_{0}-\cos\theta )+k_{3}\gamma_{OS}$$

According to Equation (18), the voltage threshold $U_{th2}$ of contact angle hysteresis of electrowetting is
(19)$$U_{th12}=\sqrt{2d_{f}\eta v_{1}\gamma_{OL}(\cos\theta_{0}-\cos\theta )/\gamma_{OS}\varepsilon_{0}\varepsilon_{1}+2d_{f}k_{3}/\varepsilon_{0}\varepsilon_{1}}$$

## 3. Model Simulation and Driving Waveform Design

Through the multi-physics field simulation software (COMSOL 5.6), the structure diagram of the electrowetting is 1.5 mm × 1.0 mm, in which the viscosity of the upper layer is 1.5 × 10^−3^ Pa·s and the lower layer is insulating ink. The section topology diagram of the electrowetting single pixel is shown in Figure 3.

The parameter settings of the electrowetting single pixel are shown in Table 1. The contact angle is 20° at zero voltage. The surface tension is 0.042 N/m. The viscosity is 5–45 cP (namely, 5 × 10^−3^–45 × 10^−3^ Pa·s). The relative permittivity is 65. And the dielectric layer thickness is 5 μm.

The electrowetting single pixel is subjected to periodic pulse signals of different peak value and different frequency and 25 V DC voltage. After the voltage is applied, the oil film shrinks. The meniscus height gradually increases and reaches the saturation state. At this time, the tension on the oil–liquid interface meets the following relationship:(20)$$P=2\gamma_{OL}/R_{c}=2\gamma_{OL}(1-\cos\theta )/h$$
where P is the pressure on the oil interface, $R_{c}$ is the radius of curvature on the oil contact surface, and h is the meniscus height of the oil film.

Since the expression of fluid pressure is
(21)P=ρgh

According to Equations (20) and (21), the expression of meniscus height h can be obtained as
(22)$$h=\sqrt{2\gamma_{OL}(1-\cos\theta )/\rho g}$$

According to the relation of the contact angle in Equation (13), the expression of the meniscus height can be obtained as
(23)$$h=\left\{\begin{array}{l} \sqrt{2\gamma_{OL}(1-\cos\theta_{0}+\varepsilon_{0}\varepsilon_{1}U^{2}/2d_{f})/\rho g},\;0\leq U<U_{th1}\\ \sqrt{2\gamma_{OL}(1-\cos\theta_{0}+\frac{\varepsilon_{0}\varepsilon_{1}U^{2}/2d_{f}+\eta v_{OS}S_{OS}/d_{OS}-k_{2}v_{OS}^{n}}{1+k_{0}k_{1}v_{OL}^{m}-k_{0}\eta v_{OL}S_{OL}/d_{OL}})/\rho g},U_{th1}\leq U<U_{th2}\\ \sqrt{2\gamma_{OL}(1-\cos\theta_{1})/\rho g},\;U\geq U_{th2} \end{array}\right.$$

Then, we solved the relevant parameters of the electrowetting model and calculated the height of the oil surface. In Figure 4, it was observed that the meniscus height of a single pixel with different viscosities changed with time under different driving voltage conditions.

As can be seen from Figure 4a–c, the meniscus height of the ink with different viscosities (0.005, 0.01, 0.015, 0.03, 0.045 Pa·s) changes when the ink is driven by the pulse signal of different amplitudes and different periods. The ink shows obvious oscillation during the applied voltage of 15 ms. Compared with Figure 4a,b, it is found that the lower the ink viscosity before 15 ms, the more obvious the motion oscillation is after applying voltage with different frequencies and the same amplitude. Although the ink with a viscosity of 0.005 Pa·s has a large amplitude of oscillation at this time, it is easier for the oil film to break with the decrease in the ink viscosity. With the increase in the viscosity, although the response speed will also decrease, the movement range of the oil film is relatively stable. And the amplitude of its movement gradually decreases. The ink with viscosity between 0.015 and 0.03 Pa·s has no obvious movement after the instantaneous oscillation, so the ink has good stability. Although the ink with viscosity of 0.045 Pa·s has no obvious oscillation, the response time is nearly 20 ms. In addition, as the frequency of the driving pulse increases, the meniscus height of the ink and the reflection brightness will also increase. By comparing Figure 4a,c, it is found that the meniscus height will increase with the increase in voltage amplitude after applying 60 Hz voltage. Then, the reflection brightness will also increase. Figure 4d shows that the meniscus height of the ink with 0.005 Pa·s viscosity changes fastest, and its instant shrinkage is the strongest at the 25 V DC voltage. However, there will be large motion oscillation. And the stabilization time is the longest; about 30 ms tends to be stable. The meniscus height of the ink with 0.045 Pa·s viscosity changes the slowest. Due to the high viscosity, the stabilization time exceeds 30 ms.

In order to view the movement process of the ink shrinkage more intuitively, Figure 5 shows a three-dimensional profile model of the meniscus height variation with time when the single pixel ink with a viscosity of 0.005 Pa·s moves at the periodic pulse voltage of 25 V peak-to-peak, with 50% duty cycle and 16 ms period. As can be seen from Figure 5a,b, the instantaneous speed of the flat ink contraction is very fast when the instantaneous voltage is applied. At 7 ms in Figure 5c, the height of the ink contraction meniscus is about to reach a high point, and the surface movement speed also decreases. In Figure 5d at 9 ms, the meniscus of the ink basically reaches a stable peak, at which time there is almost no surface movement speed.

According to the above photoelectric characteristics of the electrowetting and Equations (13), (17) and (18), as shown in Figure 6a, we have designed a composite periodic driving waveform to improve the response speed of the electrowetting and to inhibit the oil film cracking, ink backflow, and charge capture. A slope signal of 0~28 V is applied between 0 and 4 ms to prevent oil film oscillation caused by large instantaneous voltage and to inhibit oil film rupture; 28 V voltage is applied between 4 and 6 ms to make the oil film shrink and expose the substrate; the slope signal of 28~30 V is applied between 6 and 8 ms to prevent ink backflow and stabilize the oil film shrinkage; −20 V negative voltage is applied between 8 and 10 ms to eliminate the contact angle hysteresis caused by charge capture; and a −20~0 V ramp signal is applied between 10 and 14 ms to prevent ink shrinkage oscillation and return to zero voltage. Figure 6b shows the meniscus height of ink movement at the complex driving waveform. Compared with Figure 4a, it can be easily seen that there is no peak oscillation of the ink meniscus at the complex periodic driving waveform in Figure 6b. The response speed is significantly faster than that at the traditional rectangular periodic pulse driving waveform.

## 4. Experimental Testing and Results

In order to test the relationship between response time and reflection brightness of electrowetting with different viscosities at different driving voltages, we built the testing platform as shown in Figure 7. The computer is responsible for observing the movement of oil film in the pixels and calculating the aperture ratio and reflection brightness. The driving waveform required for electrowetting is used by the programmable function signal generator, the power amplifier is used to amplify the waveform power of the function signal generator, the microscope with a high-speed camera captures the shrinkage and stretching of the electrowetting, and the luminance meter is responsible for emitting standard white light and calculating the reflected brightness.

By using the D65 light source (Arges-45, Admesy, Holland) of 1200 nits, the reflectivity of yellow electrowetting pixels is 76~83%, and the reflectivity of magenta is 16.3~48.3% from closed to fully open. This was obtained by testing, collecting, and sorting the data of the photoelectric characteristics from the testing platform. According to Equations (17) and (19) and Table 1 above, the voltage threshold was calculated for the corresponding parameters for the electrowetting. A 0.5 V-step voltage was applied to the electrowetting until the voltage threshold was increased, and the change in aperture ratio was calculated. As shown in Figure 8a,b, the brightness of magenta ink increases as the voltage amplitude increases and decreases as the voltage frequency increases. This is consistent with the results of the model simulation. As shown in Figure 8c,d, the response time of yellow ink decreases as the driving voltage frequency increases. Its peak brightness increases with the increase in voltage amplitude and frequency, but the brightness changes little. The viscosity of yellow ink is lower than the magenta, and the intermolecular force between the molecules of yellow ink is smaller than that of magenta. This makes the ink shrink faster. However, it is easy to rupture the oil film and have poor contraction capacity, which leads to a higher voltage frequency and brightness but lower brightness change. Because the light transmittance of yellow ink is higher than that of magenta, the overall brightness is higher than that of magenta. Consequently, the lower the driving voltage frequency, the more effective the DC component. The higher the charge of the substrate aggregation, the greater the surface force between the two solutions, so the lower the voltage frequency applied by the magenta ink, the higher the peak brightness. In Figure 8c, due to the presence of molecular gaps in electrowetting, the electrowetting ink cannot further shrink at 25 V at 60 Hz. On the contrary, if the driving voltage is further increased, the oil film will rupture.

Figure 9 shows the 3D diagram of the photoelectric response characteristic with the magenta and yellow ink under conditions of several different frequencies and voltages. The higher the frequency of the driving voltage, the lower the brightness of the magenta oil film, and the higher the brightness of the yellow ink.

We applied the rectangular periodic pulse waveform of 25 V peak-to-peak, 50% duty cycle, 16 ms period and the composite driving waveform of Figure 6a to the multifunctional testing platform for electrowetting pixels. As shown in Figure 10, the shrinkage of single-pixel magenta ink was observed with the high-speed camera microscope. The oil film driven by the rectangular periodic pulse waveform starts to contract from the diagonal. At the same time, the instantaneous voltage causes the oil film to emit small ink droplets with high density, and its maximum aperture ratio is 52.49%. However, the oil film with the composite driving waveform contracts clockwise from the lower right corner to the upper right corner. It is obvious that the small droplets of oil film cracking are only in the lower right corner of the ink shrinkage initially, and there is no large area of oil film cracking. At the same time, its maximum aperture rate of 68.69% is larger than that driven by the rectangular periodic pulse, and its response speed is slightly faster than that driven by the rectangular periodic pulse.

## 5. Conclusions

In order to solve the problem of oil film splitting and response speed, the mathematical and physical models of contact angles were established with different viscosities at different voltages in the liquid–oil–solid three-phase system in this paper. We calculated the suitable viscosity of the oil film and its driving voltage range. A compound periodic ramp pulse was proposed for the suitable viscosity in the three-phase electrowetting system. The experimental results showed that when the viscosity of the ink is between 0.005 and 0.015 Pa·s, the higher the voltage amplitude, the lower the frequency, the higher the meniscus of the ink shrinkage, and the higher the reflection brightness. The simulation results of this model are basically consistent with the photoelectric characteristics of the magenta ink. Compared with the traditional driving waveform, experimental verification shows that the complex driving waveform can not only suppress oil film splitting, backflow, and contact angle hysteresis but also improve the response speed, the aperture ratio, and the reflection brightness of the electrowetting display.

## Figures and Tables

**Figure 1 micromachines-14-01918-f001:**
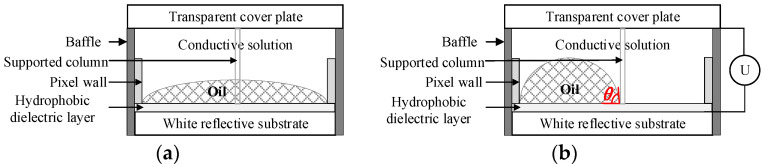
Single-pixel structure and display principle of electrowetting electronic paper. (**a**) When no voltage is applied, the pixel is turned off; (**b**) when the voltage is applied, the pixel turns on.

**Figure 2 micromachines-14-01918-f002:**
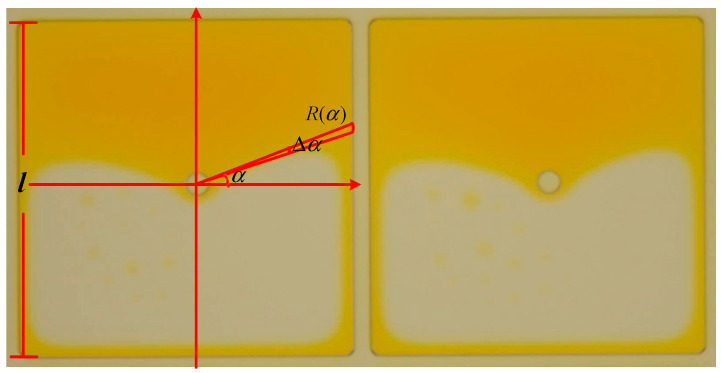
The range of aperture ratio in electrowetting pixels.

**Figure 3 micromachines-14-01918-f003:**
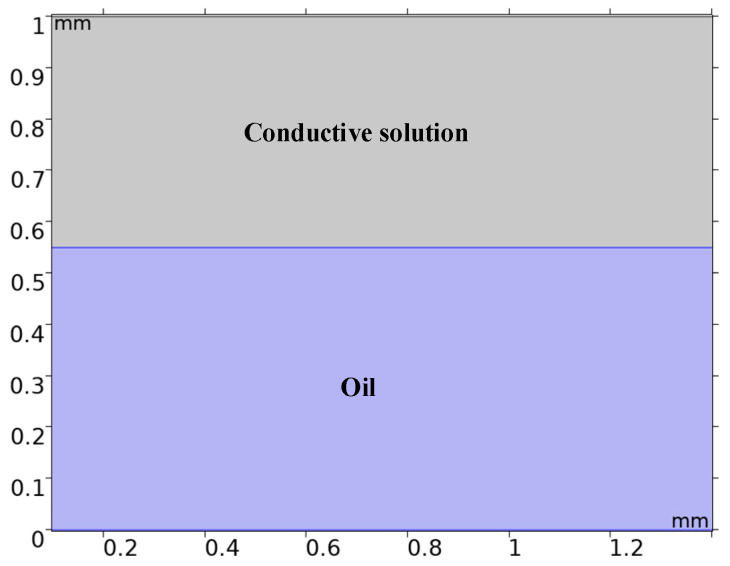
Section topology diagram of the electrowetting single pixel.

**Figure 4 micromachines-14-01918-f004:**
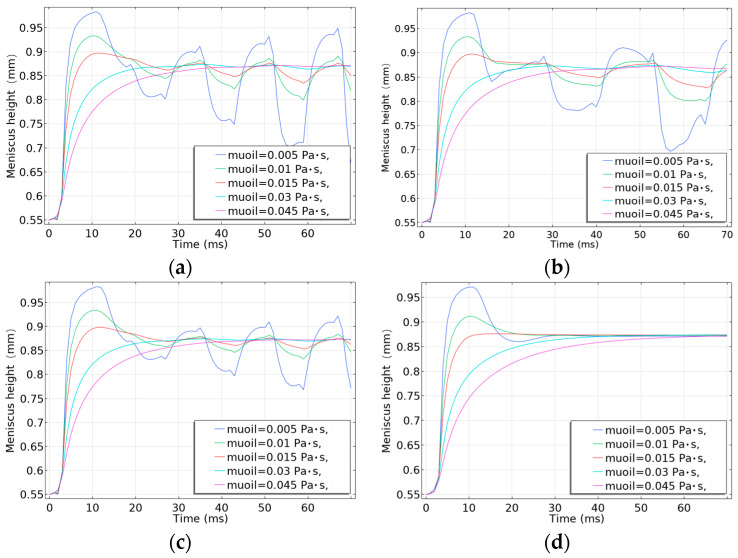
The meniscus height of a single pixel with different viscosities changed with time under different driving voltage conditions. (**a**) The peak of 25 V, the duty cycle of 50%, and the period of 16 ms; (**b**) the peak of 25 V, the duty cycle of 50%, and the period of 25 ms; (**c**) the peak of 19 V, the duty cycle of 50%, and the period of 16 ms; (**d**) 25 V DC voltage.

**Figure 5 micromachines-14-01918-f005:**
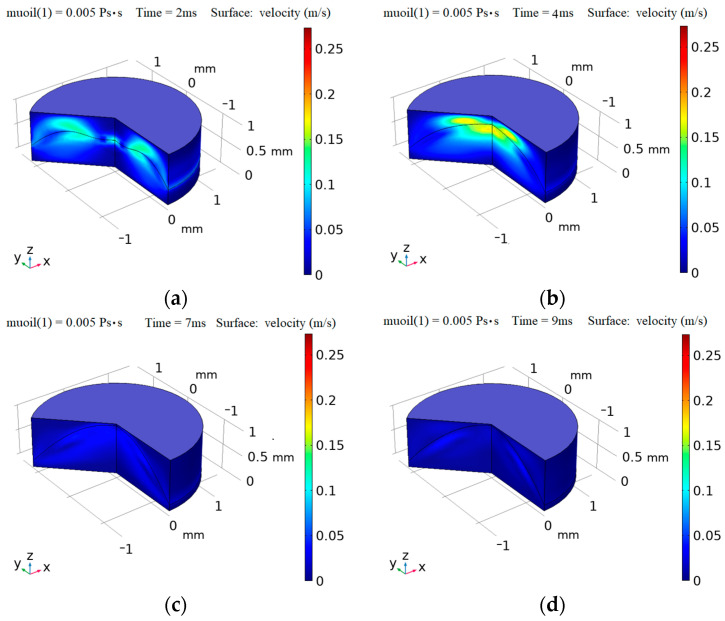
Three-dimensional profile model of the meniscus height variation with time when single pixel ink with 0.005 Pa·s viscosity moves under the periodic pulse voltage conditions of 25 V peak-to-peak value, 50% duty cycle, and 16 ms period. (**a**) 2 ms time; (**b**) 4 ms time; (**c**) 7 ms; (**d**) 9 ms.

**Figure 6 micromachines-14-01918-f006:**
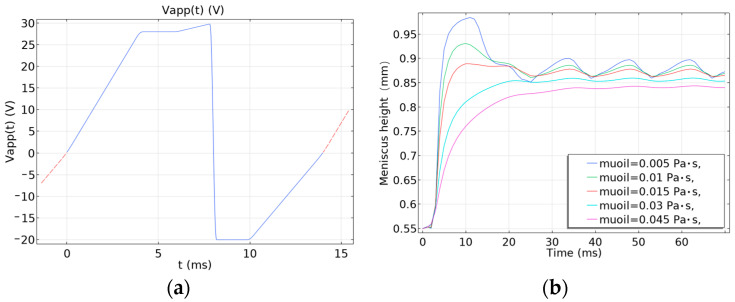
The meniscus height of ink movement at the complex driving waveform. (**a**) The composite driving waveform; (**b**) the meniscus height of ink movement.

**Figure 7 micromachines-14-01918-f007:**
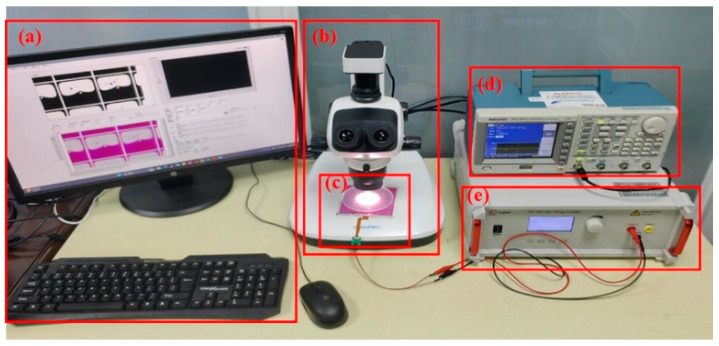
Testing platform for response time and reflection brightness of inks of different viscosities on electrowetting electronic paper pixels. (**a**) Computer; (**b**) microscope with high-speed camera and luminance meter; (**c**) electrowetting display; (**d**) programmable function signal generator; (**e**) power amplifier.

**Figure 8 micromachines-14-01918-f008:**
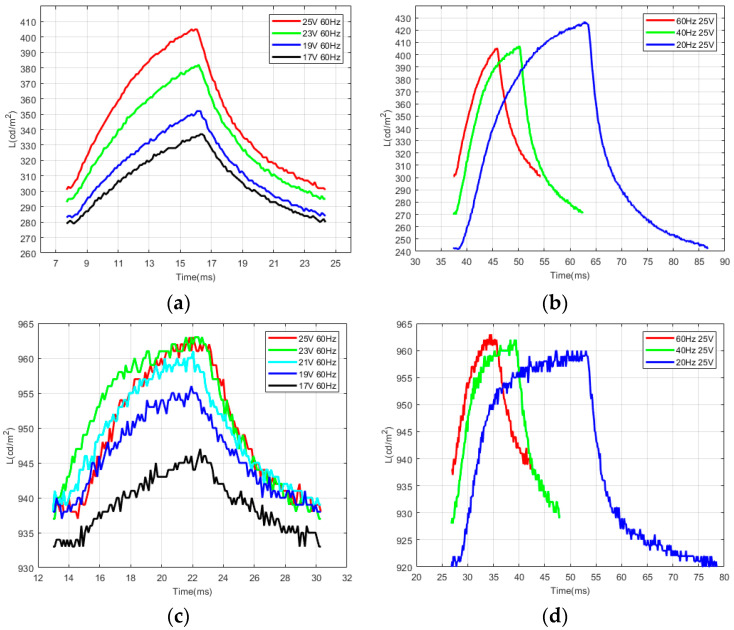
Photoelectric response characteristics of magenta and yellow ink. (**a**) Magenta ink with voltage of different amplitudes; (**b**) magenta ink with voltage of different frequencies; (**c**) yellow ink with voltage of different amplitudes; (**d**) yellow ink with voltage of different frequencies.

**Figure 9 micromachines-14-01918-f009:**
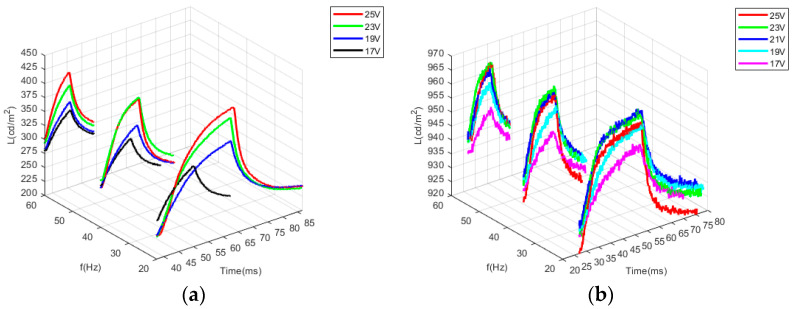
The 3D diagram of the photoelectric response characteristic with the magenta and yellow ink at several different frequencies and voltages. (**a**) Magenta ink; (**b**) yellow ink.

**Figure 10 micromachines-14-01918-f010:**
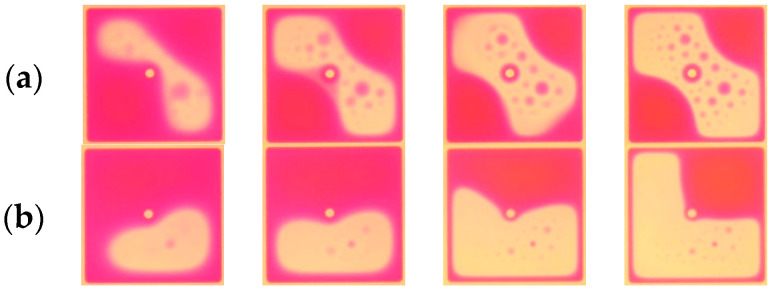
The contraction movement of magenta ink with different driving waveforms. (**a**) Driven by a rectangular periodic pulse waveform of 25 V peak to peak, 50% duty cycle, and 16 ms; (**b**) driven by complex periodic pulse waveform.

**Table 1 micromachines-14-01918-t001:** Parameter settings of electrowetting pixel.

Symbol	Quantity	Value
$\theta_{0}$	Contact angle with zero voltage (theta0)	20 deg
$\varepsilon_{0}$	Vacuum permittivity (epsr_0)	8.854187817 × 10^−12^ F/m
$\varepsilon_{1}$	Relative permittivity (epsr)	65 F/m
$d_{f}$	Dielectric thickness (d_f)	5 μm
$\gamma_{LO}$	Surface tension (gamma)	0.042 N/m
$\eta_{0}$	Viscosity of insulating oil (muoil)	5 cP

## Data Availability

The data that support the findings of this study are available from the corresponding author upon reasonable request.

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
