# Peer review of "Research on Hydrodynamic Characteristics of Electronic Paper Pixels Based on Electrowetting"

_micromachines, 2023, doi:10.3390/mi14101918_

Round 1
Reviewer 1 Report
The authors reported the new methodology to control the aperture ratio of ink/oil fluidic cells for display application. They calculated the electrowetting of the conductive solution and demonstrated it with reflection brighness. I recommend this article to published in "Micromachines" after the authors answer the below comments with major revision.
1. Line 78. Could you explain what the T Electrowetting electronic paper is? Or, it might be typo so the authors should confirm this line.
2. Line 299. The authors demonstrated the reflectance of the e-paper with the luminance meter but it need uv-vis-ir measurement. How is the reflectance/transmittance/absorbance of the cell?
3. Line 337. In figure 8(c), why is not the luminance increased at 25V with 60Hz?
Author Response
- I'm really sorry, the T Electrowetting electronic paper in line 78 is typo. Thank you for pointing it out.
- Electrowetting display is used under ambient light. Therefore, we only measured the reflection brightness of electrowetting under ambient light rather than under ultraviolet or infrared. By using the D65 light source of 1200 nits, the reflectivity of yellow electrowetting pixels is 76~83%, and the reflectivity of magenta is 16.3~48.3% from closed to fully open.
- In figure 8(c), due to the presence of molecular gaps in electrowetting, the electrowetting ink cannot further shrink at 25V with 60Hz. On the contrary, if the driving voltage is further increased, the oil film will rupture and even flip out of the pixel wall.
Reviewer 2 Report
1. Main Question Addressed by the Research:
This study aims to optimize the electrowetting display system using a proposed periodic complex pulse-driving waveform and ink with a suitable viscosity.
2. Originality and Relevance in the Field:
This research topic is both novel and pertinent to the field.
3. Contribution Compared to Other Published Material:
This research adds substantial value compared to other published material on the topic. While previous research has examined aspects of electrowetting display systems, the incorporation of a periodic complex ramp pulse driving waveform and the detailed analysis of its effects on ink response speed and pixel aperture ratio distinguishes this study as novel and significant. The reported increase in aperture ratio to 68.69% exhibits the effect of the proposed driving waveform in practice.
4. Improvements in Methodology and Further Controls:
The research methodology appears exhaustive, but there are a few considerations for enhancement. Additional information about the experimental setup and measurement methods would increase the study's openness. In addition, the authors could elucidate the selection criteria for the appropriate viscosity range and driving voltage range, highlighting their relevance to real-world applications. Introducing additional controls, such as conducting experiments with different ink types or environmental conditions, could improve the reliability of the study.
5. Consistency of Conclusions with Evidence and Arguments:
The conclusions are consistent with the presented evidence and arguments.
6. Appropriateness of References:
That is okay.
7. Comments on Tables and Figures:
That is okay.
Author Response
Thank you very much for taking the time to review this manuscript. Please find the detailed responses below and the corresponding revisions/corrections highlighted/in track changes in the re-submitted files.

Reviewer 3 Report
1. The formula derivation seems long and redundant.
2. The introduction session can't explain the significance of this work, nor give the fair literature reviews about "waveform" or "viscosity" - both concepts seem very important based on the article title.
The language and format must be improved so the reviewers can fairly offer suggestions. The language is difficult to read, Some of the texts are directly from the article templates from the Journal Office without editing.
Author Response
- Thank you very much for taking the time to review this manuscript. This article studies the relevant theories of electrowetting viscosity and driving voltage from the perspective of energy conservation. According to your request, we have properly streamlined the content of this part.
- Based on your suggestion, relevant modifications have been made to the description and significance of this work. In addition, the language and format of this article have also been revised.
Thank you very much for taking the time to review this manuscript. Please find the detailed responses below and the corresponding revisions/corrections highlighted/in track changes in the re-submitted files.
Round 2
Reviewer 1 Report
I recommend this manuscript as a formal acceptance for "Micromachines".
Author Response
We sincerely thank you for reviewing the manuscript and recognizing our work.